# Adversarial Robustness Limits via Scaling-Law and Human-Alignment Studies

**Brian R. Bartoldson** [1]   **James Diffenderfer** [1]   **Konstantinos Parasyris** [1]   **Bhavya Kailkhura** [1]

## Abstract

This paper revisits the simple, long-studied, yet still unsolved problem of making image classifiers robust to imperceptible perturbations. Taking CIFAR10 as an example, SOTA clean accuracy is about $100\%$, but SOTA robustness to $\ell_\infty$-norm bounded perturbations barely exceeds $70\%$. To understand this gap, we analyze how model size, dataset size, and synthetic data quality affect robustness by developing the first scaling laws for adversarial training. Our scaling laws reveal inefficiencies in prior art and provide actionable feedback to advance the field. For instance, we discovered that SOTA methods diverge notably from compute-optimal setups, using excess compute for their level of robustness. Leveraging a compute-efficient setup, we surpass the prior SOTA with $20\%$ ($70\%$) fewer training (inference) FLOPs. We trained various compute-efficient models, with our best achieving $74\%$ AutoAttack accuracy ($+3\%$ gain). However, our scaling laws also predict robustness slowly grows then plateaus at $90\%$: dwarfing our new SOTA by scaling is impractical, and perfect robustness is impossible. To better understand this predicted limit, we carry out a small-scale human evaluation on the AutoAttack data that fools our top-performing model. Concerningly, we estimate that human performance also plateaus near $90\%$, which we show to be attributable to $\ell_\infty$-constrained attacks' generation of invalid images not consistent with their original labels. Having characterized limiting roadblocks, we outline promising paths for future research.

## 1. Introduction

Neural networks match human performance on several tasks requiring processing of vision and text data (Achiam et al., 2023; Gemini Team, 2023). However, model performances typically falter when data is adversarially perturbed by attacks humans are robust to (Szegedy et al., 2013), calling into question model trustworthiness. Progress on this problem has been made through adversarial training (Goodfellow et al., 2014; Madry et al., 2017), but even on the "toy" dataset CIFAR10, the best robustness to $\ell_\infty$ attacks is just $71\%$ according to RobustBench (Croce et al., 2020).

The absence of a strong adversarial defense for a simple dataset like CIFAR10 is concerning because it suggests that robustifying foundation models may be impractical or even impossible (Zou et al., 2023; Bailey et al., 2023; Jain et al., 2023). Therefore, we aim to understand, *"Why is the CIFAR10 adversarial robustness problem unsolved?"*

Our experiments consider the possibility that the answer is a lack of scale used by state-of-the-art methods (Wang et al., 2023). Yet, we find that scale is not enough to achieve robustness: even with unlimited CIFAR10 data, we estimate that reaching human performance would require roughly $10^{30}$ FLOPs (see Figure 1), about 3,000 years of TF32 matrix math on 25,000 MI300 or H100 GPUs. Thus, progress on this problem requires more efficient training algorithms and improved architectures, rather than simply scaling.

Surprisingly, even if human performance could be reached, our analysis suggests that this problem will still be "unsolved": We find that human performance is around $90\%$ on AutoAttack data (see Figure 1), with the $\sim 10\%$ error induced by *invalid adversarial data* that no longer fits its label. Thus, attack formulations must be rethought to make this problem solvable; e.g., attacks should only produce valid images that abide by the original label.

Broadly, our major contributions are as follows[1]:

1. We develop scaling laws for adversarial robustness, integrating the data quality of generative models. Our scaling laws can accurately predict robustness of unseen configurations, recommend optimal resource allocations, and identify opportunities to reduce model size (raise inference speed).

2. By applying actionable guidelines from our scaling laws, compared to the prior SOTA (Wang et al., 2023), we achieve

[1]Lawrence Livermore National Laboratory. Correspondence to: Brian and Bhavya <bartoldson@llnl.gov, kailkhura1@llnl.gov>.

*Proceedings of the 41st International Conference on Machine Learning*, Vienna, Austria. PMLR 235, 2024. Copyright 2024 by the author(s).

[1]Appendix A provides a detailed list of our contributions and their implications, with links to relevant sections to aid navigation.

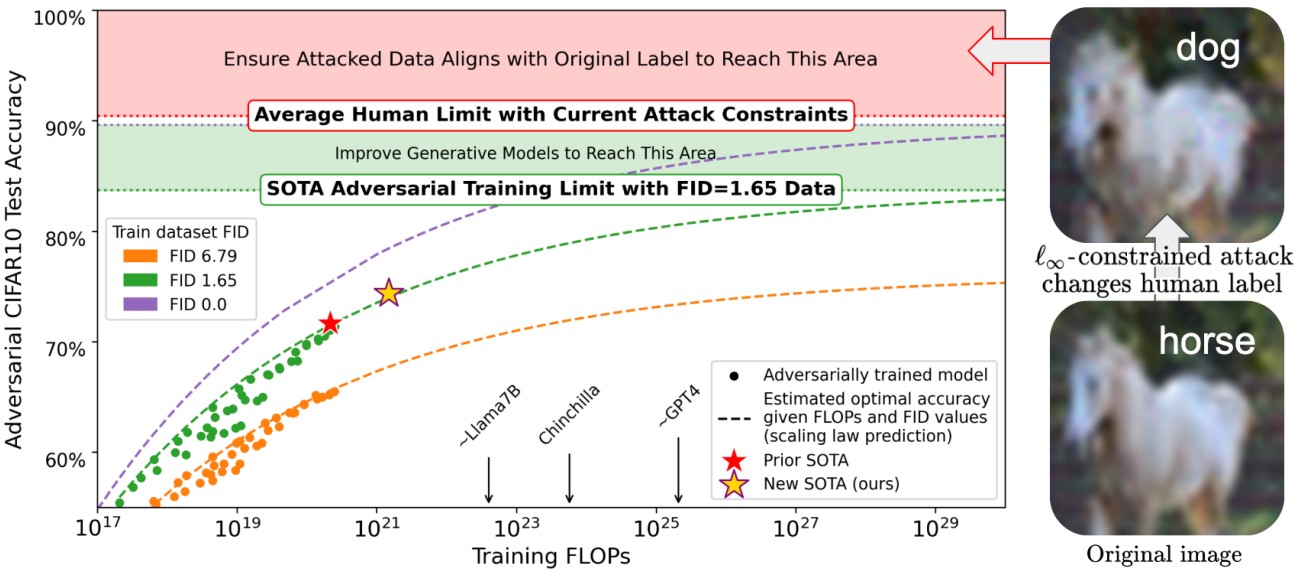

*Figure 1.* **SOTA techniques cannot solve the CIFAR10 adversarial robustness problem, even with infinite compute.** (**left**) Scaling laws predict that performance asymptotes near 90%. We produce this estimate using Approach 2 (Section 3.2.2), a scaling law approach we designed that can model the effect of FID, allowing approximation of performance when training on a hypothetical FID=0.0 synthetic dataset of arbitrarily large size. Note that lowering FID raises dataset quality. We learned the parameters of Approach 2's parametric model of testing performance by fitting to the performances that resulted from training various NNs on various datasets; the depicted dots show a subset of these performances. (**right**) A CIFAR10 test image is adversarially perturbed by AutoAttack, causing humans to disagree with the ground-truth label "horse". Consistent with scaling law asymptotes, our small-scale human study (Section G) estimates that the original label is no longer an appropriate label for roughly 10% of adversarial data. Benchmarking that uses the original label after it ceases to be a good fit for the image will suggest that NN robustness is further from being "solved" (human-level) than it truly is.

the same robustness level with 20% fewer FLOPs, enable a +1% AutoAttack gain with a $3\times$ smaller model trained with the same compute budget, and set a new SOTA (+3% AutoAttack accuracy) using a larger compute budget.

3. Our analysis suggests that advancements in generative models, efficient algorithms, and model architectures will help address this problem to some extent but will not solve it completely. We find that the $\ell_\infty$ attack formulation is flawed and creates *invalid adversarial data* that humans also misclassify. Therefore, solving this problem requires fixing the attack formulation to account for image validity.

## 2. Related Work

**Adversarial Training Trends** It is common practice to adversarially train on CIFAR10 data by modifying training data via a 10-step attack (Rebuffi et al., 2021; Sehwag et al., 2021; Wang et al., 2023; Peng et al., 2023), making an adversarial training iteration $9\times$ as expensive as a regular one (see Appendix B). Further adding to the costs of adversarial training, recent work find that larger models (Singh et al., 2023; Huang et al., 2023) and larger datasets improve adversarial robustness. Indeed, synthetic CIFAR10 datasets with

as many as 50M samples have been used to boost adversarial training's effectiveness (Rebuffi et al., 2021; Gowal et al., 2021; Sehwag et al., 2021; Wang et al., 2023). Our SOTA model trains on 300M unique synthetic CIFAR10 data points, requiring over $10^{21}$ training FLOPs (Figure 1).

Aware of the cost needed just to make CIFAR10 models moderately robust, recent research on safeguarding LLMs and vision language models (VLMs) against attacks discusses the potential difficulty or even futility of robustifying foundation models (Zou et al., 2023; Bailey et al., 2023; Jain et al., 2023; Sun et al., 2024). Before incorporating any adversarial training or robustness measures, the training cost of such models is already high—Figure 1 shows the FLOPs costs of training: GPT4 (Achiam et al., 2023), a closed model with a rough cost estimate (Epoch, 2023); Llama7b (Touvron et al., 2023), the cost of which we estimate by applying the FLOPs = $6ND$ heuristic of Kaplan et al. (2020) to 1T tokens and 6.7B parameters; and Chinchilla (Hoffmann et al., 2022). Notably, such costs have been tempered by use of compute-efficient training settings derived from scaling laws fit to LLM performances (Kaplan et al., 2020). We consider the possibility that similar scaling laws could help address the inefficiency of adversarial training.

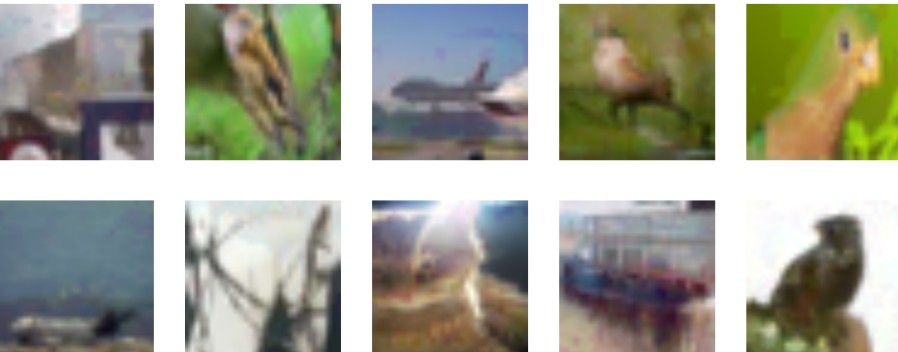

*Figure 2.* **Subtle attacks transfer to humans, showing brittleness in human visual perception**. Try to classify these adversarial images that fool our SOTA neural net as CIFAR10 classes (airplane, automobile, bird, cat, deer, dog, frog, horse, ship, and truck). Next, see Figure 21 for the clean counterparts of these images to see if you were also fooled. Section G further discusses such "invalid" data.

Given the robustness benefits of adversarial training with large synthetic datasets (Rebuffi et al., 2021; Gowal et al., 2021) and with improved synthetic data quality as measured by FID (Wang et al., 2023), our scaling law approach generalizes prior scaling laws by modeling the effect of synthetic data quality. As a result, our scaling law predictions change as training dataset FID changes (see Figure 1), which is not the case in smaller concurrent studies of the scalability of adversarial training (Debenedetti et al., 2023). Notably, prior work incorporated data quality into language model scaling laws by decreasing the contribution of duplicative training data (Muennighoff et al., 2023), whereas we do not train on duplicate training data (except where noted) and model quality using FID. When data quality is sufficiently high, we find that efficient adversarial training scales the model size and dataset size at roughly the same rate, consistent with Chinchilla's scaling laws (Hoffmann et al., 2022).

**Adversarial Examples That Fool Humans**   In Figures 1 and 2, we show that AutoAttack (Croce & Hein, 2020) on our SOTA NN produces adversarial data that tricks both humans and NNs. Prior work shows humans are also fooled by adversarial data with higher resolutions and other attack constraints (Elsayed et al., 2018; Guo et al., 2022; Veerabadran et al., 2023; Gaziv et al., 2023). For instance, small-$\ell_2$ norm adversarial perturbations, which are often discussed as mostly unnoticeable by humans, can fool humans on high-resolution ImageNet data (Gaziv et al., 2023). Gaziv et al. (2023) find that humans are more easily tricked by adversarial data if the attacked network is first robustified.

Models not trained to be robust can also be used to generate adversarial data that fool humans, though humans are only fooled by such images when given limited time to classify them (Elsayed et al., 2018). In Gaziv et al. (2023) and in this work, adversarial data is generated using robust networks,

and humans may not deduce the correct category even with extra time. The ability to trick humans using images that trick robust neural networks may not be surprising given the overlap in frequencies relied upon by each system to make classifications (Subramanian et al., 2023),[2] the potential for a shared mechanism for robustness in humans and machines (Harrington & Deza, 2021), and the susceptibility of biological, primate neurons to adversarial attacks (Guo et al., 2022). However, we identify that such human susceptibility causes benchmarking of robustified networks (Croce et al., 2020) to produce adversarial perturbations that induce changes in human labeling: this means benchmarking does not return a fraction of "gold-standard" human performance and instead poses the robustness problem as one that is unsolvable by networks with human-level robustness. Supporting this unsolvability on CIFAR10, the human limits we find (Figure 1's red line) match our scaling laws' asymptotes (e.g., the FID=0 limit is the top of the shaded green area in Figure 1).

## 3. Scaling Laws for Adversarial Robustness

In this section, we discuss the methodology for our scaling laws. We start by training a range of models, varying model size, dataset size, and synthetic dataset quality. We then use these models' test data performances to fit empirical estimators of how the loss should vary with these factors.

### 3.1. Background and Methodology

**Adversarial Robustness**   Both adversarial attacks and defenses can be understood using the following formulation:

$$\min_\theta \rho(\theta), \text{where } \rho(\theta) = \mathbb{E}_{(x,y)\sim\mathcal{D}}\left[\max_{\delta\in\mathcal{S}} L(f_\theta(x+\delta), y)\right] \quad (1)$$

---

[2]Humans actually use a subset of the frequencies used by neural networks. See Appendix C for more detailed discussion.

and here $\mathcal{D}$ is the data distribution over image-label pairs $(x, y)$, $f$ is the neural network parameterized by $\theta$, $L$ is the cross-entropy loss, and the set of allowable perturbation vectors $\delta$ that may be applied to $x$ is $\mathcal{S} = \{\delta \mid \|\delta\|_\infty \leq 8/255\}$. The adversarially attacked image $x + \delta$ created in the maximization step of Equation 1 can be found via $t = 1$ (Goodfellow et al., 2014) or $t > 1$ (Madry et al., 2017) steps of projected gradient descent (PGD) on the negative loss:

$$x^t = x + \delta^t, \text{ where}$$
$$\delta^t = \text{proj}_{\mathcal{S}} \left( \delta^{t-1} + \alpha \, \text{sign}(\nabla_{\delta^{t-1}} L(f_\theta(x^{t-1}), y)) \right), \quad (2)$$

$\delta^0$ is a random initial perturbation in the set $\mathcal{S}$, and $\text{proj}_{\mathcal{S}}(\delta)$ projects the candidate perturbation $\delta$ onto the set $\mathcal{S}$ so that no pixel changes by more than $8/255$. Defenses against such attacks can be improved by attacking data at training time, i.e., by optimizing $\rho(\theta)$ or "adversarially training". Performances on adversarial and regular datasets can be balanced by using the TRADES loss (Zhang et al., 2019).

**Adversarial Training Settings**   We follow the adversarial training approach of Wang et al. (2023), a leading CIFAR10 approach on the RobustBench leaderboard (Croce et al., 2020). Specifically, we use: 10 PGD steps to adversarially attack images at training time with step size $\alpha = 2/255$; label smoothing (Szegedy et al., 2016) 0.1; synthetic CIFAR10 data from generative models in our training datasets; the TRADES loss (Zhang et al., 2019) with $\beta = 5$; weight averaging (Izmailov et al., 2018) with decay rate $\tau = 0.995$; SGD optimization with Nesterov momentum (Nesterov, 1983) set to 0.9 and weight decay $5 \times 10^{-4}$; and a cyclic learning rate schedule with cosine annealing (Smith & Topin, 2017). The learning rate and batch size we use varies based on dataset size according to optimal settings found by hyperparameter search: datasets with 10M or fewer samples use batch size 1024, otherwise batch size is 2048; the learning rate is 0.3 for datasets with 10M or fewer samples, 0.2 for larger datasets with up to 200M samples, and 0.1 for datasets with 300M samples (see Appendix D).

**Training Datasets**   We build training datasets from three different synthetic data generators. When training models used to fit scaling laws, datasets contain synthetic CIFAR10 data of one specific quality (FID), and we generate enough data to only see each example once during training, allowing our scaling laws to reflect the benefit of adding a unique data point. Note that this mirrors how scaling laws in the LLM literature are learned; e.g. Hoffmann et al. (2022) train on each data source for one epoch, with some minor exceptions (e.g., Wikipedia is seen $\sim 3\times$). Our approach has similar, rare exceptions (see Appendix E).

We generate synthetic data via the elucidating diffusion model (EDM) (Karras et al., 2022), the Poisson Flow Generative Model plus plus (PFGM++) (Xu et al., 2023), and

Table 1. Our synthetic data generators and their image qualities.

| | **Synthetic Data Generators** | | | | | | |
|---|---|---|---|---|---|---|---|
| Type: | EDM | EDM | EDM | EDM | EDM | PFGM++ | DG |
| Steps: | 5 | 6 | 7 | 10 | 20 | 18 | 20 |
| FID | 35.54 | 14.26 | 6.79 | 2.48 | 1.82 | 1.76 | 1.65 |

Table 2. Each model we use and its parameter count $N$ in millions.

| | **WideResNet Configurations** | | | | | | | |
|---|---|---|---|---|---|---|---|---|
| Depth: | 28 | 40 | 82 | 28 | 58 | 82 | 70 | 82 |
| Width: | 4 | 4 | 4 | 12 | 12 | 12 | 16 | 16 |
| $N \times 10^{-6}$ | 6 | 9 | 20 | 53 | 122 | 178 | 267 | 316 |

Discriminator Guidance (DG) (Kim et al., 2022). We use each generator and a variety of diffusion steps to create 7 training dataset generators with various image qualities (see Table 1), which we measure with FID (Heusel et al., 2017).

For each training dataset generator, we generate 100 million (EDM-7, EDM-20, PFGM++, DG) or 30 million (EDM-{5, 6, 10}) unique samples. For generators with 100 million samples, we train models on 5, 10, 30, 70, and 100 million samples, corresponding to traditional CIFAR10 epoch counts of 100, 200, 600, 1400, and 2000. The other generators are only used to train on 5,10, and 30 million samples.

**Models**   We train the WideResNet models (Zagoruyko & Komodakis, 2016) shown in Table 2. Peng et al. (2023) boost the prior SOTA AutoAttack performance of Wang et al. (2023), 70.69% to 71.07%, by modifying their WideResNet architecture, but we opt for WideResNet as it is a widely-used architecture for this problem (Croce et al., 2020). All models use SiLU activations (Hendrycks & Gimpel, 2016).

**Evaluation**   In Section 3.3, we report accuracy on AutoAttack (Croce & Hein, 2020), which performs 4 unique attacks (including modified versions of PGD) and is tracked by the RobustBench leaderboard (Croce et al., 2020). In our scaling law plots, we report either model loss or accuracy, which are computed on the CIFAR10 test set after its images were adversarially attacked using 40 PGD steps on the model's CW loss (Carlini & Wagner, 2016). Note that the reported adversarial loss is the TRADES loss with $\beta = 5$, mixing vanilla and adversarial performances. Also note that the adversarial accuracies that result from this PGD attack tend to be about 1% higher than those resulting from AutoAttack (Wang et al., 2023) but are significantly cheaper to obtain. We derive FLOPs values via the approach in Appendix B.

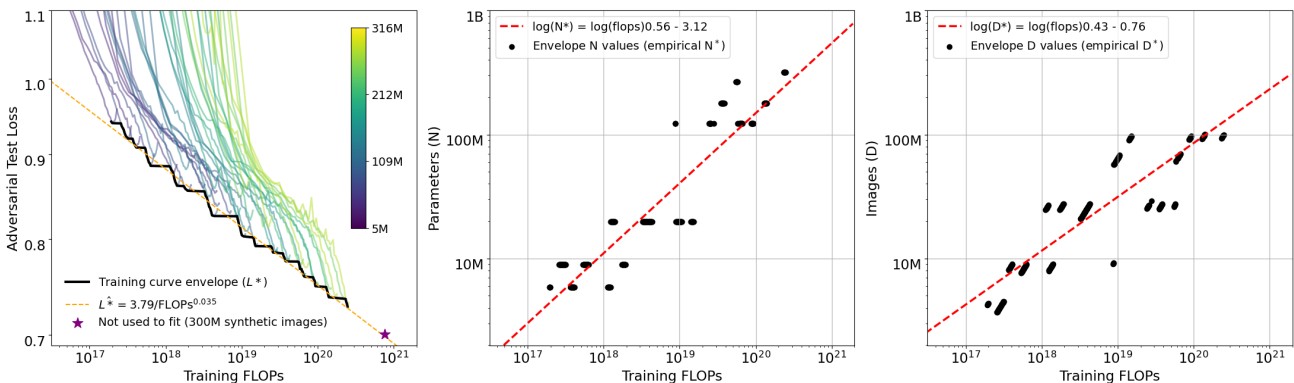

*Figure 3.* **Approach 1 fits to the envelope of training curves.** This adversarial robustness scaling law approach does not model the effect of training dataset FID on performance, so it is only fit to results associated with one training dataset. For example, in the **left** plot, we show the learning curves of our models trained on PFGM++ synthetic data. When training, we vary $N$ between 6M and 316M and $D$ between 5M and 100M. Given the optimal $N, D$ combination at each FLOP level (the "envelope"), we estimate how the optimal $N$, $N^*$ (**middle**), and optimal $D$, $D^*$ (**right**), changes as a function of FLOPs via log-space regressions on points sampled from the envelope.

## 3.2. Deriving Scaling Laws

In this section, we explore three parametric forms for our scaling laws to fit the data generated from our large-scale experiments. Next, we use our scaling laws to recommend: a) the optimal resource allocation (i.e., model and dataset size) at various training FLOPs values, allowing robustness to be maximized given a budget; and b) the optimal return on investment, i.e., the best robust loss/accuracy that can be achieved for a given compute budget. Finally, we also derive the tradeoff between model size and training compute overhead, showing that there are opportunities to reduce model size (and improve inference speed).

### 3.2.1. APPROACH 1: ENVELOPE OF TRAINING CURVES

Our first approach to understanding the scaling behavior of adversarial training is inspired by the "Approach 1" used in the Chinchilla scaling study (Hoffmann et al., 2022). Specifically, given a collection of 8 models with parameter counts $N$, each trained on up to 5 different dataset sizes $D$, we find the lowest loss at each training FLOP value – these are the points on the black envelope line in Figure 3. We fit a power law to predict the lowest loss $L^*$ at these compute budgets. We also identify the $N, D$ combination with the lowest loss at each training FLOP value. Using these optimal configurations, we fit power laws to predict the optimal $N, D$ from the training FLOPs budget, obtaining relations of the form $N^* \propto \text{FLOPs}^a$ and $D^* \propto \text{FLOPs}^b$.

We repeat Approach 1 for each data generator, and we find similar results for data generators with similar qualities (see Appendix F.1 for more on Approach 1). However, as we improve data quality—e.g., moving from EDM-5 to EDM-6 to EDM-20—we notice two changes: (1) the minimum

loss at each FLOP level improves; and (2) the exponents $a, b$ move closer towards 0.5, the values they took in Hoffmann et al. (2022). For example, $a = 0.73, b = 0.27$ for EDM-5, but the PFGM++ results shown in Figure 3 have $a = 0.56, b = 0.43$, which is consistent with the idea that lower quality data is less beneficial (Wang et al., 2023). Moreover, the fact that improving data quality raises $b$ and lowers the loss suggests that a rigorous understanding of scaling adversarial training must account for data quality, consistent with results on non-synthetic, non-adversarial CIFAR10 scaling (Sorscher et al., 2022) and motivating our Approaches 2 and 3. Indeed, unlike Approach 1, Approaches 2 and 3 can fit to training runs that used various FIDs, and predict performance as a function of FID.

### 3.2.2. APPROACH 2: PARAMETRIC LOSS WITH DATA QUALITY TERMS

Approach 2 addresses the inability of Approach 1 to model two important limiting cases: (1) what happens when $N, D$ tend toward infinity; and (2) what happens when $N, D$ are at practical levels but the training dataset has unlimited high-quality CIFAR10 examples (i.e., an FID=0 synthetic dataset). For example, as shown in Figure 1, Approach 2 can predict performance on an FID=0 training dataset and the asymptotic performance as a function of FID.[3]

To achieve this, Approach 2 parameterizes the loss with the functional form used in "Approach 3" of Hoffmann et al. (2022), but it modifies two constants related to the ability of $D$ to affect asymptotic performance so that they

---

[3]In Figure 1, Approach 2 predicts optimal losses, then we convert these losses to accuracies (see Appendix F.2.2 for details).

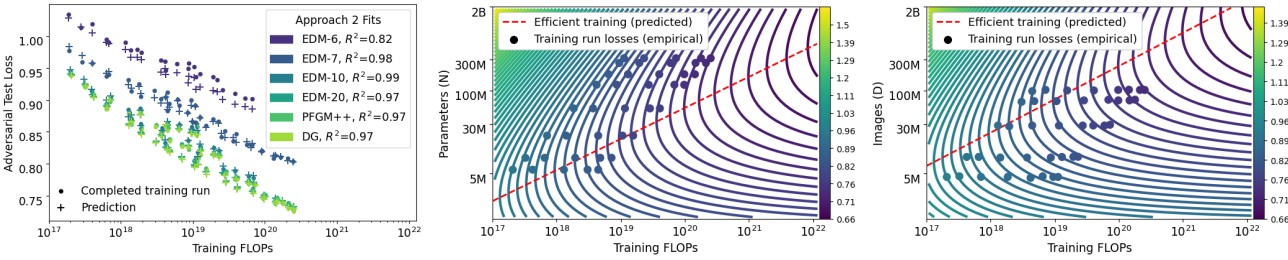

*Figure 4.* **Approach 2 models the effect of synthetic dataset quality.** Approach 2 can fit to training runs that used datasets with differing FIDs by modeling FID's effect on performance. (**left**) Equation 3 fits runs that used different models and dataset sizes/qualities. EDM-6 and PFGM++ were not used to learn the fit's parameters, but are plotted to show extrapolation performance of the fit. We plot isoLoss contours (regions with the same loss predicted by our scaling laws; higher losses are brighter) and the curves for $N^*$ (**middle**) and $D^*$ (**right**), the training settings predicted to be compute-optimal at a FLOPs budget for a synthetic dataset with DG20's quality.

are functions of data quality as shown in Equation 3:

$$L(N, D, \text{Quality}) = \frac{A}{N^\alpha} + \frac{B'}{D^\beta} + E',$$
$$E' = \exp(\log(E) + \log(1 + \text{Quality}^{-1})\epsilon), \quad (3)$$
$$B' = \exp(\log(B) + \log(1 + \text{Quality}^{-1})\zeta),$$

where $\epsilon$ and $\zeta$ model how changes in data quality affect the irreducible loss $E$ and the portion of the loss reducible by dataset growth $B$, respectively, and $\text{Quality} = \frac{1}{\text{FID}}$. Note that with infinitely high synthetic data quality (i.e., FID=0 data), Equation 3 simplifies to $L = A/N^\alpha + B/D^\beta + E$.

To learn the parameters of Equation 3, we first fit all but $\epsilon, \zeta$ to 40 training runs on the DG dataset (8 models trained on DG's 5 sizes). We then hold the learned parameters constant and fit $\epsilon, \zeta$ to models trained on three datasets of varying data quality (EDM-7, EDM-10, EDM-20). We use the L-BFGS optimization approach described in Hoffmann et al. (2022) for both fitting procedures (see Appendix F.2 for more details on the fitting procedure). In Figure 4, we show how the fitted parameters accurately predict loss for the dataset used to fit the base parameters (DG), the datasets used to fit the quality parameters (EDM-7, EDM-10, EDM-20), and two held out datasets (EDM-6, PFGM++).

Figure 4 also shows the optimal $N, D$ for each training FLOPs level on the DG dataset. We derive these curves by analytically minimizing Equation 3 with respect to $N$ ($D$) after replacing $D$ ($N$) using the constraint FLOPs $\approx$ $7822ND$, rather than the constraint FLOPs $\approx 6ND$ used for non-adversarial LLM training (Kaplan et al., 2020; Hoffmann et al., 2022). Our constraint is found to never have more than 1% relative error across all observations of our training runs (more than 9000 observations covering 233 different model-dataset combinations). This minimization gives $N^*$ ($D^*$) as a power law function of FLOPs and the exponent $a$ ($b$): $N^* \propto \text{FLOPs}^a$ ($D^* \propto \text{FLOPs}^b$). Specifically, we have the following:

$$N^* = G\left(\frac{\text{FLOPs}}{7822}\right)^a, \quad D^* = G^{-1}\left(\frac{\text{FLOPs}}{7822}\right)^b, \quad (4)$$

where $G = \left(\frac{\alpha A}{\beta B}\right)^{\frac{1}{\beta+\alpha}}, a = \frac{\beta}{\beta+\alpha}, b = \frac{\alpha}{\beta+\alpha}$.

### 3.2.3. APPROACH 3: PARAMETRIC LOSS MODELING DATA QUALITY AND ITS EFFECT ON OVERFITTING

While Approach 2 models the effect of synthetic data quality, it is possible that its functional form does not capture the complex relationship between data quality and loss scaling. Indeed, Approach 2 models quality rather simply, and empirically we find that the fit of this simple power law form is poor when the fitting includes the lowest quality dataset (EDM-5, FID=35.54). Accordingly, we test the validity of the scaling laws derived by Approach 2 by using a more complex functional form that can simultaneously model the loss on all datasets accurately.

Our Approach 3 is inspired by Equation 4.1 of Kaplan et al. (2020), which models how increasing $N$ causes overfitting (and loss plateauing) earlier as $D$ is reduced. Here, rather than modeling this "data size bottleneck" that restricts the benefit of increasing $N$, we model a "data quality bottleneck" that restricts the benefit of increasing $D$. Compared to Kaplan et al. (2020), we use a similar functional form but couple $D$ with Quality rather than with $N$:

$$L(N, D, \text{Quality}) = \frac{A}{N^\alpha} + \left(\frac{B}{D} + \left(\frac{Q}{\text{Quality}}\right)^{(\kappa/\beta)}\right)^\beta + E',$$
$$\text{where } E' = E + \log(1 + \text{Quality}^{-1})\epsilon. \quad (5)$$

Like Approach 2, our Approach 3 simplifies at FID=0 to $L = A/N^\alpha + B/D^\beta + E$, assuming that the constant $B$ is set to $B^\beta$. We fit Equation 5 using all datasets and models, finding $L$ is accurately predicted due to the ability to model overfitting as a function of data quality (see Figure 5). Since Equation 5 lacks a simple derivative with respect to

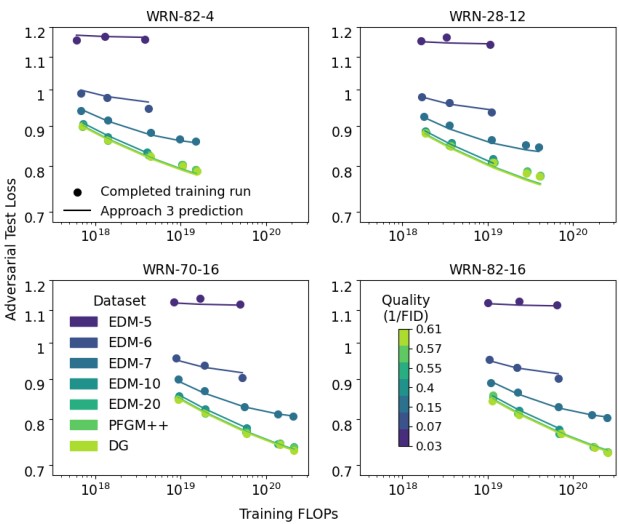

*Figure 5.* **Approach 3 fits to models trained on data of any FID.** Our third approach can account for overfitting happening sooner on lower quality datasets, allowing it to simultaneously model training runs from all of our datasets, even those with very low qualities. As a result, Approach 3's learned parameters account for more training run results than other approaches' parameters do.

*Table 3.* **All scaling law approaches suggest similar conclusions.** Comparison of $a$, $b$, $L^*$ at $10^{25}$ FLOPs, and $E$ (if modeled).

| | PFGM++ (FID=1.76) | | | | Hypothetical (FID=0) | | | |
|---|---|---|---|---|---|---|---|---|
| Approach | $a$ | $b$ | $L^*$ | $E'$ | $a$ | $b$ | $L^*$ | $E$ |
| 1 | 0.56 | 0.43 | 0.50 | – | – | – | – | – |
| 2 | 0.48 | 0.52 | 0.61 | 0.56 | 0.48 | 0.52 | 0.53 | 0.48 |
| 3 | – | – | 0.63 | 0.55 | 0.47 | 0.53 | 0.56 | 0.52 |

$N$, $D$ at nonzero FIDs, we numerically solve for its $N^*$, $D^*$. Appendix F.3 further discusses Approach 3 and its L-BFGS fitting, and Figure 12 extends Figure 5 to show all 8 models.

Ultimately, we obtain similar results with each approach (Table 3 compares them), bolstering support for our scaling-law-related conclusions. Also, Figure 13 shows that Approaches 2 and 3 have nearly identical $N^*$ predictions for FID $= 0$ (and shows all approaches' $N^*$ for other FIDs).

### 3.3. Putting Scaling Laws into Practice

#### 3.3.1. COMPUTE-EFFICIENT ADVERSARIAL TRAINING

We use our scaling laws to predict compute-optimal training settings ($N^*$, $D^*$) at a variety of compute budgets. As shown in Figure 6, we train models according to these settings and find AutoAttack accuracy surpasses the prior SOTA's (Wang et al., 2023). Thus, past inability to reach

higher robustness levels is partially addressed via more efficient training settings. Further scaling also helps robustness.

We highlight the following results (see Figure 6):

**(1)** A new SOTA: 73.71% AutoAttack and 93.68% clean accuracy via a WideResNet-94-16 and 500M samples.

**(2)** A model predicted to be compute-optimal at the prior SOTA's training FLOPs budget (WideResNet-82-8, 79M parameters) beats the prior SOTA (WideResNet-70-16, 267M parameters) by 1% AutoAttack, reaching 71.59% AutoAttack and 93.11% clean accuracy, while being over $3\times$ more FLOP and parameter efficient.

**(3)** A model trained entirely on synthetic data outperforms the prior SOTA, attaining 71.7% AutoAttack accuracy and 93.27% clean accuracy.

Given the benefit of training on higher quality data observed in all of our approaches (e.g., Figure 1), we boost dataset quality for these training runs by mixing CIFAR10 data in with the synthetic data at a rate of 1 CIFAR10 example for every 7 synthetic examples. Such mixing was also done to obtain the prior SOTA (Wang et al., 2023). However, note that we also train one model entirely on synthetic data.

In Appendix F.4, we explore Figure 6 and Table 3 more thoroughly. For example, we discuss the variation in $N^*$ predictions and why Approach 3 provides a bridge between the other two approaches. Of note, while $N^*$ predictions vary by approach, all of our scaling laws agree that the prior SOTA was trained with too large a model, which is just outside the compute-efficient region in Figure 6.

#### 3.3.2. TRADING OFF TRAINING EFFICIENCY FOR INFERENCE EFFICIENCY

In real-world scenarios, it is often helpful to deviate from compute-optimal settings by using extra resources to train a smaller model longer, gaining memory and inference efficiency without harming performance. To guide this decision making process, we study the tradeoff between model size and compute overhead (Kaplan et al., 2020; Vries, 2023).

Given a desired model size that is $\omega_N\times$ as large as the compute-optimal model size $N^*$, we can find the multiple $\omega_D$ of the optimal dataset size $D^*$ needed to maintain the expected loss of the optimal model and dataset sizes. In particular, we ensure no change in our Approach 2 scaling law predicted loss by choosing $\omega_D$ as follows:

$$E' + \frac{A}{N^{*\alpha}} + \frac{B'}{D^{*\beta}} = E' + \frac{A}{(\omega_N N^*)^\alpha} + \frac{B'}{(\omega_D D^*)^\beta},$$

$$\text{where } \omega_D = \left(1 - (\omega_N^{-\alpha} - 1)\frac{AN^{*-\alpha}}{B'D^{*-\beta}}\right)^{-\frac{1}{\beta}}$$

(6)

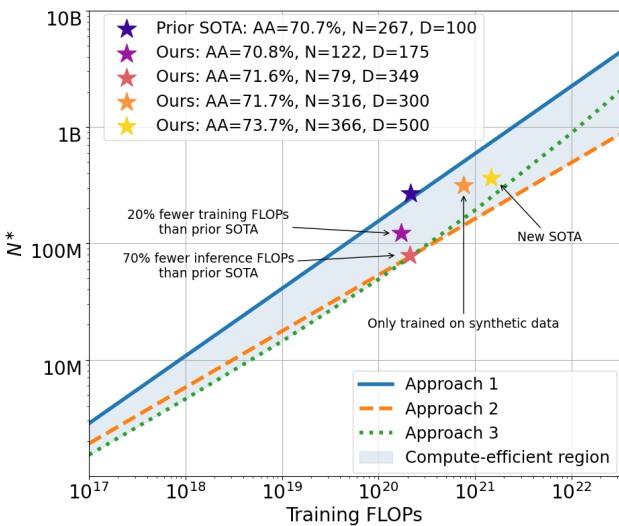

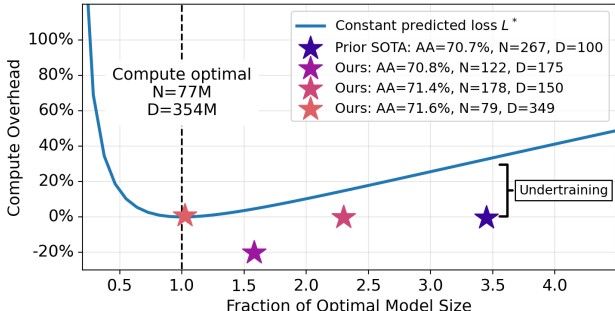

*Figure 7.* **Train smaller models on larger datasets.** Given the FLOPs budget of the prior SOTA, we find the optimal model size using Approach 2 (assuming DG-20 data), its predicted loss $L^*$, and the expected compute overhead incurred by training other model sizes to $L^*$. The prior SOTA is not expected to reach $L^*$ due to its small training dataset—consistent with this, we train smaller models on more data and find AutoAttack accuracy increases.

*Figure 6.* **Training settings estimated to be compute-efficient by our scaling laws are state-of-the-art, validating our scaling law fits.** We find $N^*$ for the DG-20 dataset for Approaches 1–3 and train models using values of $N$ in between at least two approaches' predicted values for $N^*$ (see the "compute-efficient region"). The trained models are more efficient and robust than the prior SOTA.

and $E'$ and $B'$ assume a particular dataset quality. Given new data and model sizes, the "compute overhead" is the increase in training FLOPs as a percentage of the training FLOPs needed for the optimal model and dataset sizes.

As can be seen in Figure 7, there exists a substantial region where the model size can be reduced with minimal compute overhead. For example, reducing the model size by half ($\omega_N = 0.5$) results in only a ∼10% increase in computational requirements, demonstrating an efficient tradeoff between model compactness and training cost. We also show that, given the prior SOTA's training FLOPs (fixing compute overhead to 0%), larger models are predicted to require increasingly more compute to reach the performance of the compute-optimal model, consistent with the displayed AutoAttack performances falling as model size increases.

## 4. Discussion

By deriving scaling laws for CIFAR10 adversarial robustness, we showed that compute-efficient configurations (i.e., model and dataset sizes), better synthetic data generators, and more compute can all help improve adversarial robustness. Indeed, we improve on the prior SOTA by 3%, reaching 73.71% AutoAttack accuracy. However, we also find that there are asymptotic limits to performance: it appears that some adversarial data will never be correctly classified. To address this surprising finding, we evaluate human performance on the adversarial data that our SOTA NN mis-

classifies, and we find that humans fail to classify much of this data too (Appendix G). That is, when applied to NNs with SOTA robustness, existing attack models that bound attacks using $\ell_p$-norm constraints generate adversarial data that is *invalid*. This finding is consistent with prior works that criticize the ability of the $\ell_p$ norm to quantify perceptual similarity (Sharif et al., 2018; Tversky, 1977). Invalid data prevents benchmarking from returning a fraction of human-level robustness. Accordingly, we argue for designing an attack formulation that takes image validity into account.

### 4.1. Limitations and Future Directions

We sought to clarify adversarial robustness limits suggested by the **unsolved state** of a well-studied, small-scale adversarial robustness problem: $\ell_\infty$-norm-constrained AutoAttack on CIFAR10 (Croce et al., 2020). In doing so, despite the large scale of our experiments (over $10^{22}$ total FLOPs and hundreds of training runs), we made observations that need to be tested more broadly. For instance, prior work suggests that invalid data arises on higher resolution datasets like ImageNet (Gaziv et al., 2023), but the extent to which invalid data affects performance on other robustness benchmarks requires further study. Future work can extend ours by exploring other impacts of invalid data on the robustness problem. We focused primarily on the effect of invalid data that arises during benchmarking, but our preliminary results show that invalid data can be created by training attacks too (Appendix H.4). We also made observations about robustness scaling behavior that should be tested with other datasets and models: studying WideResNets, we found that $N$ and $D$ should be scaled at roughly the same rate when synthetic data quality is high – it will be interesting to examine this more generally (e.g. with ViTs on ImageNet).

## Impact Statement

This paper presents work whose goal is to advance the field of Machine Learning. There are many potential societal consequences of our work, none which we feel must be specifically highlighted here. Our organization's IRB deemed our work to have no need for an IRB review.

## Acknowledgements

We thank Nicholas Carlini, Maksym Andriushchenko, Vikash Sehwag, and Pin-Yu Chen for helpful comments. We thank Youngsoo Choi for facilitating our human evaluation. This work was performed under the auspices of the U.S. Department of Energy by Lawrence Livermore National Laboratory under Contract DE-AC52-07NA27344 and LDRD Program Project No. 23-ERD-030 (LLNL-CONF-860190).

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

# A. Outline of Our Contributions and Their Implications

Here we present a more detailed list of our contributions and their implications.

Our contributions are as follows:

- We derive the first scaling laws for CIFAR10 adversarial robustness that take synthetic data quality into account (Sections 3.2.2 and 3.2.3).

- Using data generated from large-scale experimentation (238 training runs and over $10^{22}$ FLOPs), we show that our scaling laws can accurately model and predict robustness performances across various unseen model sizes, dataset sizes, and synthetic data qualities (Section 3).

- We compute: (a) the optimal resource allocation (i.e., model and dataset size) at various training FLOPs values, allowing robustness to be maximized given a budget; and (b) the optimal return on investment, i.e., the best robust loss/accuracy that can be achieved for a given compute budget (Equation 4 and Figure 6 for (a), and Figure 1 for (b)).

- We also derive the tradeoff between model size and training compute overhead, showing that there are vast opportunities to reduce model size (improve inference speed) while actually decreasing training compute overhead (Equation 6 and Figure 7).

- We carry out a small-scale human study to gain a deeper understanding of the performance bottleneck of current adversarial training methodologies (Section G).

- Additionally, we are the first to perform a single-epoch (i.e., seeing each image only once), synthetic-data-only adversarial training run that outperforms the prior state-of-the-art, which relied on using non-synthetic data (Figure 6).

Via our scaling law and human studies, we provided actionable insights to advance the field of adversarial robustness:

- With the compute-optimal settings from our scaling laws, we showed that the prior SOTA adversarial training run (Wang et al., 2023) used a model too large for its training FLOPs budget, using more compute than necessary to achieve its robustness level, and we matched this prior robustness peak with 20% fewer training FLOPs (Figures 6 and 7).

- We trained several new models in a compute-efficient manner to make a significant leap in SOTA robustness – our best model beats the prior SOTA by 3% AutoAttack accuracy (Figures 1 and 6).

- A model predicted to be exactly compute-optimal at the prior SOTA's FLOPs budget (Wang et al., 2023) outperformed the prior SOTA by 1% AutoAttack while being $3\times$ smaller (FLOPs and parameters), boosting training performance and inference efficiency (Figure 7).

- We found that "scaling up compute" alone is not an effective strategy to achieve a human robustness level, as our scaling laws predict that current approaches will need thousands of years of state-of-the-art supercomputer time to reach this level (Figure 1).

- We also found that human performance itself on this task is far from perfect: existing attack formulations (relying on an $\ell_\infty$-norm bound to ensure attacks preserve the original label) concerningly allow generation of invalid images that do not fit their original label (Figures 1, 14, and 15).

Some broader implications of our work for the adversarial robustness area are as follows:

- Advances on the adversarial robustness problem will require the design of more efficient training algorithms and improved architectures, rather than simply scaling (Section 4).

- Attack formulations must be rethought to solve this problem; e.g., attacks should only produce valid images that contain the information needed to assign the original label (Section G).

- Robustifying models closer to human levels makes them greater threats for misuse, as they can be combined even with naive attack approaches ($\ell_\infty$-norm-bounded attacks) to create adversarial examples that transfer to humans, fooling them and creating potential security concerns (Section G).

# B. Counting the FLOPs of Adversarial Training

We use the PyTorch (Paszke et al., 2019) flop counter to find the FLOPs consumed by adversarial training. We find that our adversarial training loop requires $27\times$ the FLOPs required to complete 1 forward pass, whereas regular training is typically estimated to require about $3\times$ the forward pass FLOPs. The added cost of adversarial training is primarily due to the 10 PGD iterations used to adversarially perturb images at training time. The following explains this in detail.

### B.1. Why is adversarial training $9\times$ as expensive per iteration?

We clarify why an adversarial training iteration, using 10 PGD steps and the TRADES loss, requires $\sim9\times$ the FLOPs of a regular training iteration. First, we note that a typical training iteration uses 1 forward pass and 1 backward pass, which is $\sim2\times$ as FLOP-intensive as a forward pass. We approximate this as 3 forward passes of compute. Next, we show that a typical adversarial training iteration requires 27 forward passes of compute.

Creating an adversarial image for the TRADES loss requires a PGD attack that only computes the gradient with respect to the image (not the weights), making the backward pass for each step of this attack only $\sim1\times$ as FLOP-intensive as a forward pass. Thus, for 10 PGD steps, we use $\sim20$ forward passes of compute. This number rises to 21 because a forward pass on the clean data is used to obtain the clean-image logits used in the PGD attack.

Two additional forward passes are used to compute the TRADES loss, which compares the output of the model on the attacked and clean data (a more efficient implementation could save a forward pass by reusing the clean data output needed by the PGD attack). Finally, there is a backward pass used to compute the gradient of the TRADES loss. This backward pass is $\sim2\times$ as large as a typical backward pass because the batch size is effectively doubled via the loss function's use of model outputs from two different inputs. Thus, given a PGD-attacked image, the TRADES loss requires 6 forward passes of compute, making the entire adversarial training iteration require about $\sim27$ forward passes of compute.

# C. Additional Discussion of Adversarial Data That Fools Humans

Elsayed et al. (2018) generate adversarial examples by attacking CNNs and show that human accuracy falls on these perturbed images when the humans are only briefly exposed to the images and asked to quickly classify the image. While Elsayed et al. (2018) find that human predictions are generally not harmed by their study's small adversarial perturbations when the humans are given sufficient time to make their choice, we find that adversarial data created by common attack approaches do fool humans, making that adversarial data's human label misaligned with the label used to compute accuracy.

Our finding that perturbations that cause artificial network misclassification often also cause human misclassification may not be surprising. Adversarial robustness comparisons of humans and artificial neural networks are further explored by Subramanian et al. (2023), who show that humans make classifications using a narrower range of frequencies within the range of frequencies relied upon by artificial networks for classification decisions. Thus, perturbations that fool neural networks may act on frequencies that humans rely on for classification, potentially explaining the results of our human-NN comparison study. This is consistent with human comments after the study: e.g., "the attack was able to manipulate the image in some way that it made it very confusing (either changing semantics, blurring part of the object/class into the background, etc.)".

Guo et al. (2022) show the susceptibility of biological, primate neurons to adversarial attacks, indicating that use of adversarial training that aligns artificial neural network performance with human performance may not produce models impervious to adversarial attack. Our work is consistent with the argument that making artificial models more human-like will not solve the adversarial robustness problem, as we observe that images that fool state-of-the-art neural networks also fool humans. Indeed, our results show that measures of progress on the adversarial robustness problem may be too pessimistic – i.e., by showing that humans often fail to classify adversarially perturbed CIFAR10 images, we clarify that neural networks are closer to human-level ("gold-standard") performance than previously thought.

# D. Hyperparameter Study

Prior to training models used to fit our scaling laws, we conducted a hyperparameter study to understand how to set key hyperparameters as a function of model size and dataset size. A sample of our findings from this study is shown in Figure 8. We use such results to support the settings we discuss using in our main text, which we reiterate here: datasets with 10M or fewer samples use batch size 1024, otherwise batch size is 2048; the learning rate is 0.3 for datasets with 10M or fewer samples, 0.2 for larger datasets with up to 200M samples, and 0.1 for datasets with 300M samples.

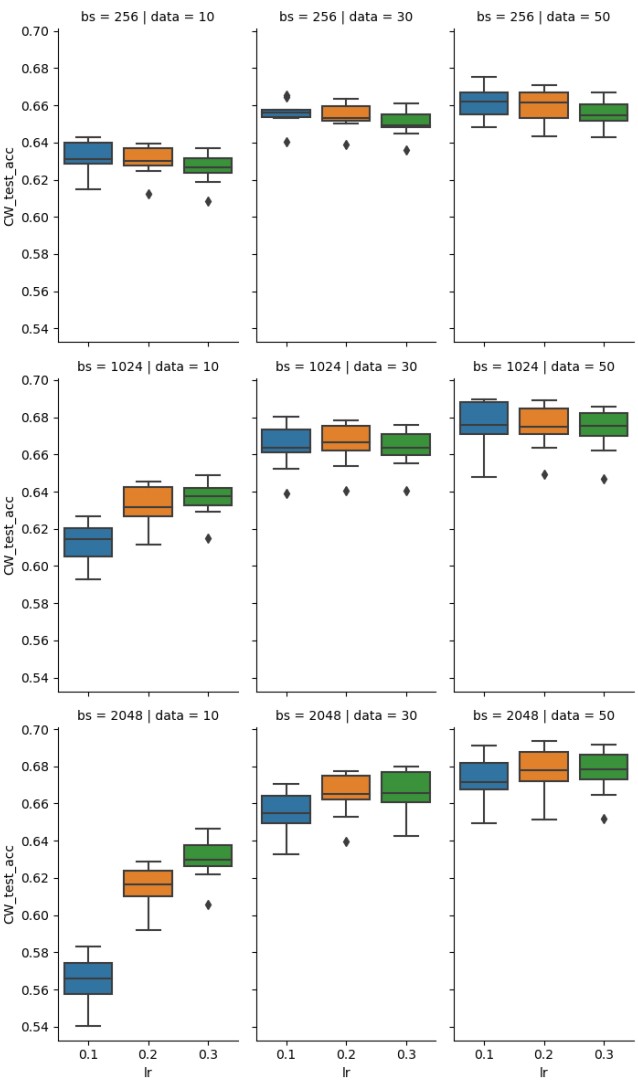

*Figure 8.* **Hyperparameter study.** We trained a variety of preliminary models on different dataset sizes ("data" in the plot, shown in millions) in order to understand how to set key hyperparameters like batch size ("bs" in the plot) and learning rate ("lr" in the plot) prior to training models for our scaling laws. Settings with higher test accuracy after a PGD attack on the CW loss ("CW_test_acc" in the plot) were chosen for our scaling law training runs.

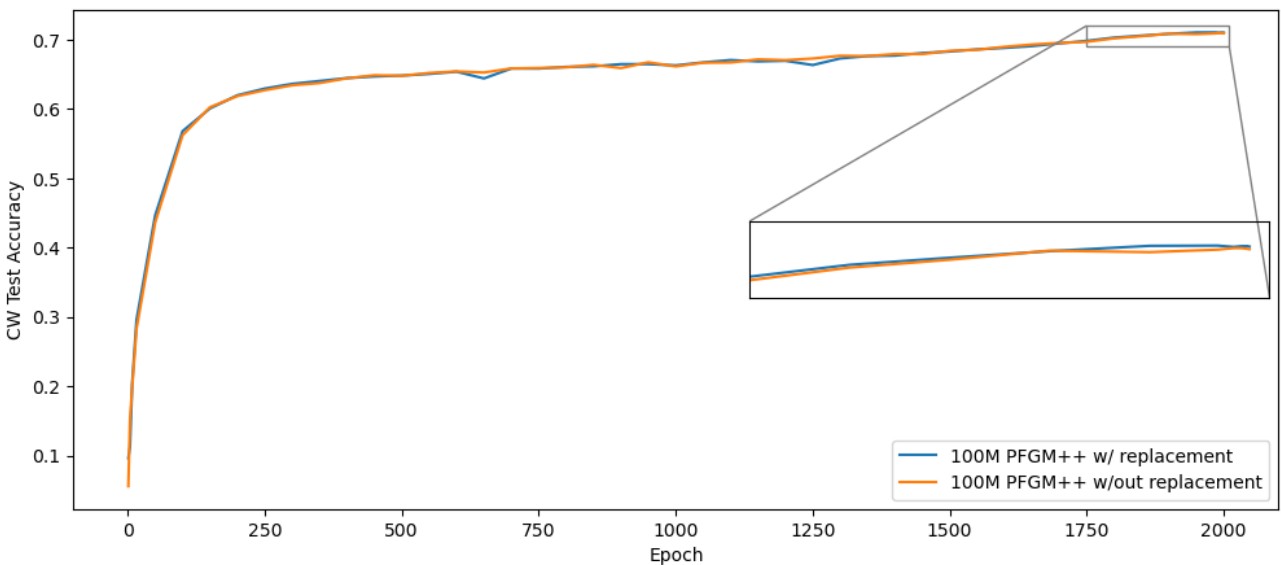

*Figure 9.* **Sampling with and without replacement are similar.** Infrequent resampling of the same sample that sometimes occurs when training models used to fit our scaling laws has no notable effect on performance.

## E. Training Dataset Sampling

When training on synthetic data, we aim to see each sample a single time. However, we sometimes see a specific sample more than once during a training run for three reasons:

1. Our data sampler uses a constant batch size for each training step, causing the actual number of samples seen per epoch to be slightly above the 50,000 typical for CIFAR10 training data. Our training epochs calculation is made assuming 50,000 samples are seen each epoch, so we start to reuse synthetic samples seen at the start of training to finish the final few epochs. For example, assume we see 51,200 samples per epoch: if training on 5M samples, we would train for 100 epochs and revisit $100 \times (51{,}200 - 50{,}000) = 120{,}000$ samples. All other samples would be seen once during training.

2. Training the SOTA robustness model requires 500 million high-quality synthetic examples, and we train on EDM-20 for 1.5 "epochs", DG for 1.5 "epochs", and PFGM++ for 2 "epochs" (reaching a total of 500 million examples). In this paragraph, "epochs" refers to passes through the full 100,000,000 element datasets. In most of the training runs discussed in Section 3.3, we also mix in non-synthetic CIFAR10 data, so the actual number of full passes through each of these synthetic datasets is slightly lower.

3. When training models used for our scaling laws, we sampled images without replacement (in a random order) but did not reload the data sampler's state when interrupted jobs resumed. This leads to some amount of sampling *with* replacement, which was the strategy used by Wang et al. (2023). In particular, when training was interrupted by a job time limit (every 12 hours), our training would resume with a new (random) data sampler state, causing some previously seen examples to be resampled. We show that this form of infrequent resampling causes no notable difference relative to training without replacement (by reloading the data sampler state) in Figure 9. The models we train in Section 3.3 reload the data sampler state to ensure sampling without replacement.

Similar to Wang et al. (2023), we see non-synthetic CIFAR10 examples multiple times during training when mixing together original CIFAR10 and synthetic data in our training runs not used for scaling laws (Section 3.3).

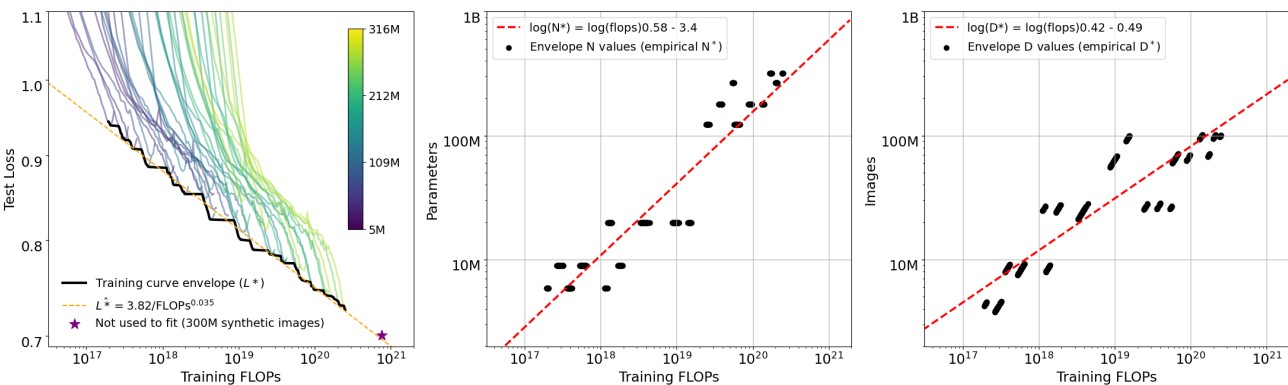

*Figure 10.* **Approach 1: Envelope of Training Curves.** Our findings in Figure 3 are stable when the training dataset is changed to DG.

## F. Adversarial Scaling Laws: Additional Results and Details

### F.1. Approach 1

Given empirically observed $N^*, D^*$, the points on the envelope in Figure 3, Approach 1 uses regressions in log space to find the optimal $N^*, D^*$ as power law functions of FLOPs. The results of these regressions are shown in Figure 3.

The query FLOPs values used to collect $N^*, D^*$ are obtained by taking a set $\mathcal{F}$ of 1000 logarithmically spaced values of training FLOPs, then restricting to the subset $\mathcal{Q} = \{f_i \in \mathcal{F} \mid \nexists f_j \in \mathcal{F} \text{ such that } (Loss_j < Loss_i) \wedge (f_j < f_i)\}$. The restricted query points are those that reflect loss improvements when the training FLOPs budget increases; i.e., the restriction removes upticks in the envelope at regions with an insufficiently dense sampling of training dataset sizes, areas of Figure 3 where the envelope is flat. The optimal loss at each query training FLOP value produces the envelope seen in the black line of Figure 3.

When using other datasets with high quality, we find similar results to those shown in Figure 3. For instance, Figure 10 shows similar results are obtained when using DG instead of PFGM++.

### F.2. Approach 2

#### F.2.1. FITTING

We use L-BFGS to learn the parameters of Equation 3, reproduced below for convenience:

$$L(N, D) = \frac{A}{N^\alpha} + \frac{B'}{D^\beta} + E',$$
$$E' = \exp(\log(E) + \log(1 + \text{Quality}^{-1})\epsilon),$$
$$B' = \exp(\log(B) + \log(1 + \text{Quality}^{-1})\zeta).$$

Our approach largely follows that of Hoffmann et al. (2022), with the exception that we also learn the values of $\epsilon, \zeta$. L-BFGS optimizes the Huber loss with $\delta = 0.001$. The observed error we feed into the Huber loss is computed in the following way. We begin with candidate values for $a, b, e, \alpha, \beta, \epsilon, \zeta$. We define $a = \log(A)$. We transform the logs of $B, E$ ($b, e$ at FID=0) according to the data quality parameter; for example: $b = \log(B) + \log(1 + \text{Quality}^{-1})\zeta$. We then compute the log of the predicted loss: $\log(\hat{L}) = \log\_sum\_exp([a - \alpha\log(N), b - \beta\log(D), e])$. Finally, the error used in the Huber loss is $\log(\hat{L}) - \log(L)$.

When learning $A, B, E, \alpha, \beta$, we initialize L-BFGS with each element in the Cartesian product of the following initial points, ultimately selecting the values associated with the minimum achieved Huber loss. $a : [0, 1, 2, 5, 10]$, $b : [0, 1, 2, 5, 10]$, $e : [-1, -.5, 0, .5, 1]$ $\alpha : [0, 0.1, 0.25, 0.5, 1]$, $\beta : [0, 0.1, 0.25, 0.5, 1]$.

When learning $\epsilon, \zeta$, we initialize L-BFGS with each element in the Cartesian product of the following initial points, ultimately selecting the values associated with the minimum achieved Huber loss. $\zeta : [-.3, -.15, .15, .3]$, $\epsilon : [0.01, 0.1, 0.2]$.

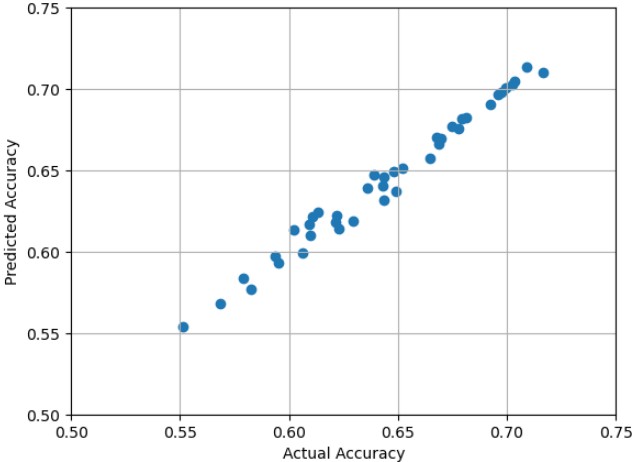

*Figure 11.* **Predicting accuracy from loss.** We create an accuracy predictor to facilitate conversion of predicted loss values to predicted accuracy values.

We obtain the following values:

$A = 6.69$, $B = 9.89$, $E = 0.48$, $\alpha = 0.24$, $\beta = 0.23$, $\epsilon = 0.16$, $\zeta = -0.28$.

Note that we fit Equation 3 to all of the data specified in Section 3.2.2 with one exception: we do not fit to "small" models trained on "large" datasets. The specific restriction is that we drop training runs where the following conditions are all true: model size is less than 100M, data size is greater than 10M, FID is less than 10. We discuss the reason for this restriction in Section F.3, and below we show that removing this restriction does not cause our Approach 2 fitted values to change much:

$A = 7.6$, $B = 9.70$, $E = 0.49$, $\alpha = 0.25$, $\beta = 0.23$, $\epsilon = 0.14$, $\zeta = -0.25$.

### F.2.2. LOSS-ACCURACY REGRESSION

In order to create Figure 1, which predicts adversarial accuracy values, we convert our scaling law's loss predictions to accuracy predictions. To do this, we regress accuracy on loss using observations of models trained on the DG dataset, restricting to the subset of observations at which models obtain their minimum loss. Using this regression-based-predictor results in good agreement between predicted and actual values ($R^2 = 0.98$), as shown in Figure 11. The learned slope is $-0.7496$, the learned intercept is $1.2575$.

### F.3. Approach 3

### F.3.1. FITTING

We use L-BFGS to learn the parameters of Equation 5, reproduced below for convenience:

$$L(N, D, \text{Quality}) = \frac{A}{N^\alpha} + \left( \frac{B}{D} + \left( \frac{Q}{\text{Quality}} \right)^{(\kappa/\beta)} \right)^\beta + E',$$

$$E' = E + \log(1 + \text{Quality}^{-1})\epsilon.$$

The fitting procedure largely mirrors the one used for Approach 2 described in Section F.2, and we describe notable differences in this paragraph. When fitting Approach 3, in addition to using results from all of the datasets listed in Table 1, we also use the performance of one model trained on the 300M synthetic images from our three best generators (EDM-20, PFGM++, DG), which is visible in the bottom right plot of Figure 12. When learning $A, B, E, Q, \alpha, \beta, \kappa, \epsilon$, we initialize L-BFGS with each element in the Cartesian product of the following initial points, ultimately selecting the values associated with the minimum achieved Huber loss. $A : [5, 6, 7]$, $B : [6500, 7000, 7500]$, $E : [.6, .5]$, $Q := [0.01, 0.5]$, $\alpha : [0.1, .2, .3]$, $\beta : [0.1, .2, .3]$, $\kappa : [.8, .6]$, $\epsilon : [.01]$.

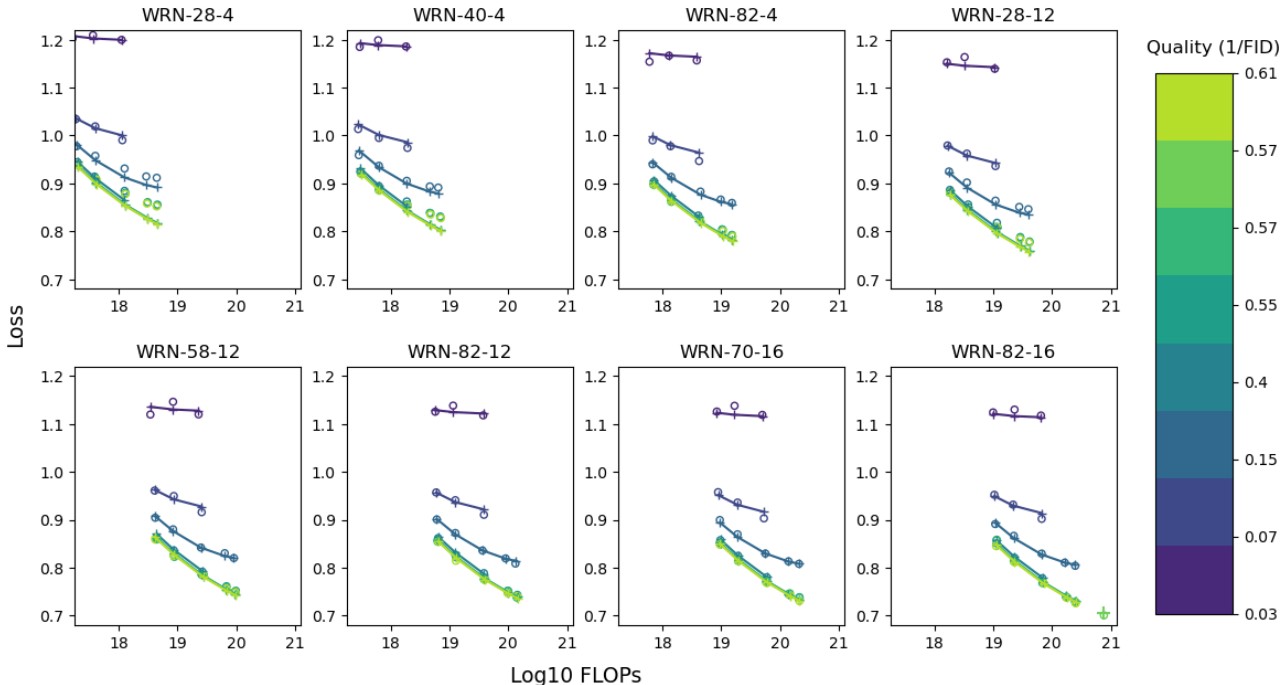

*Figure 12.* **Fit of Approach 3 on all models.** This is a version of Figure 5 that shows the fit of Approach 3 to the other 4 models used to learn our scaling laws. Note that the same scaling law parameters are used in all 8 plots, demonstrating Approach 3's ability to simultaneously capture the effects of model, dataset, and quality scaling.

The specific values we learn are the following:

$A = 6.0,\ B = 7000.0,\ E = 0.52,\ Q = 0.007,\ \alpha = 0.24,\ \beta = 0.22,\ \kappa = 0.6,\ \epsilon = 0.04.$

In Figure 12, we show that these values produce good fits for all models, datasets, and qualities. Note that at FID=0, Equations 3 and 5 have similar forms and their values can be compared. The main exception to this is that $B$ in Approach 2 should be compared to $B^\beta = 6.85$ in Approach 3.

We fit Equation 5 to all of the data specified in Section 3.2.3 with one exception: we do not fit to "small" models trained on "large" datasets. The specific restriction is that we drop training runs where the following conditions are all true: model size is less than 100M, data size is greater than 10M, FID is less than 10. We impose this restriction because small models display underfitting on larger, higher quality datasets. For an example of this, see the top left plot of Figure 12, where WRN-28-4 losses are above the predicted values at larger FLOP values (i.e., larger dataset sizes). Modeling this underfitting behavior with our scaling law is possible, but we opted instead to learn a scaling law that applies to models that are not trained on far too much data given their size, as current adversarial training works typically do not enter this regime (using large model sizes and smaller dataset sizes).

### F.3.2. COMPUTING OPTIMAL N, D

Unlike Equation 3, the minimum of Equation 5 does not have a simple analytical form. Accordingly, we find $N^*$ and $D^*$ using a numerical optimization approach (SciPy's 'fsolve' routine).

### F.4. Compute-Efficient Adversarial Training

**Models and datasets used**  Here we provide additional model and dataset details for the results shown in Figures 6 and 7.

We train WideResNet-82-8 (79M parameters), WideResNet-58-12 (122M parameters), WideResNet-82-12 (178M parameters), WideResNet-82-16 (316M parameters), and WideResNet-94-16 (366M parameters).

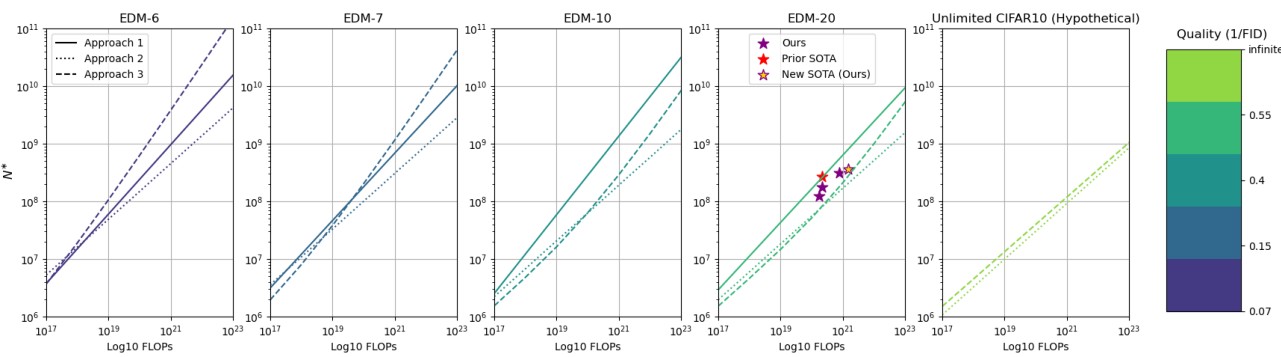

*Figure 13.* **N* comparisons.** As data quality improves, our three scaling law approaches tend to predict smaller $N^*$. A version of the fourth plot is shown in the main text in Figure 6.

The 500M sample dataset used for our SOTA model (shown in Figure 6) is constructed from 300M unique samples as described in Appendix E. The model trained on 150M samples uses 50M samples from each of our highest quality synthetic datasets (EDM-20, PFGM++, DG). For the model trained on 175M samples, we use 25M samples from this collection of 150M samples twice. The model trained on 349M samples used 100M samples from each of our highest quality synthetic datasets, seeing 49M samples twice. The model trained on 300M samples used 100M samples from each of our highest quality synthetic datasets and had no CIFAR10 data mixed in.

**Comparing the Three Scaling Law Approaches**  In Table 3 and Figure 6, we see that Approach 1 favors scaling the model size more aggressively than Approach 2 ($a = 0.56$ vs. $a = 0.48$). Approach 3 represents a balance between the two approaches, shifting from suggesting the scaling of the model and dataset sizes at similar rates (at low training FLOPs values) to suggesting that additional compute should be primarily allocated toward increasing the model size due to its anticipating overfitting to the dataset of limited quality (at high training FLOPs values). Note that at the highest quality data (FID=0), Approach 3 suggests a constant $a, b$ similar to Approach 2's, as shown in the rightmost plot of Figure 13.

All of our approaches suggest that the prior SOTA is trained on too little data given its model size. Thus, a central message of our results is to train smaller models on more data, consistent with the idea that robustness has a large sample complexity (Schmidt et al., 2018).

**Compute-Efficient Region**  In Figures 6 and 7, we use the DG-20 dataset to create the compute-efficient region, assuming that our training data will have FID $\approx 1.65$. Due to the mixing in of non-synthetic CIFAR10 training data, the data quality available to the models shown in Figures 6 and 7 could be slightly better than FID=1.65, which could mean the actual $N^*$ is even lower due to the tendency of higher quality data to raise $b$.

## G. Small-Scale Human Performance Study

The irreducible error captured by our scaling laws (Table 3) indicates that, even with unlimited compute, the CIFAR10 $\ell_\infty$ AutoAttack benchmark problem cannot be solved. Here, we investigate whether this irreducible error is attributable to a flaw in our scaling laws or a fundamental intractability of achieving robustness to $\ell_p$-norm-bounded attacks.

One possibility, supported by recent work (Gaziv et al., 2023), is that adversarial attacks on highly robust models produce adversarial images containing no clear object or an object of the wrong class, which we call *invalid adversarial data*. If attacks during benchmarking create such invalid data, then irreducible error (or a limit to benchmark performance that is below 100%) would be expected because both humans and neural networks would by definition fail to classify such images correctly (except by chance).

Hence, in an effort to understand the cause of the irreducible adversarial error in our scaling laws, we perform a small-scale human performance evaluation on the 2629 adversarial images that were incorrectly classified by our SOTA model (with 73.71% AutoAttack accuracy). Through this study, we **(1)** *Identify that invalid adversarial data is used when benchmarking CIFAR10 robustness with $\|\delta\|_\infty = 8/255$ attacks*: Human performance suggests that 727/2629 images are "invalid".

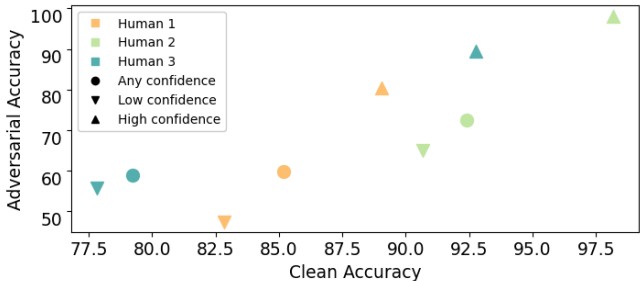

*Figure 14.* **Adversarial accuracy vs. clean accuracy by human and confidence.** Classification accuracy of each human user on the clean and adversarial versions of the 2629 images incorrectly classified by our SOTA NN. Accuracy is provided for collections of predictions where users indicated any, low, or high confidence.

This finding characterizes the extent to which existing benchmarking does not track the fraction of gold-standard "human performance" achieved by the NN. In other words, it suggests the need for an improved attack formulation that takes data validity into account. **(2)** *Propose a revised benchmarking scheme* to address the nonzero irreducible error of the $l_\infty$ attack formulation: human-identification of invalid images due to class deception or ambiguity, then removing invalid data from benchmarking. We also encourage more research in designing attack formulations that take image validity into account.

### G.1. Evaluating Human Performance

To understand human performance on benchmarking data, we first saved the adversarial and clean (pre-attack) versions of the 2629 CIFAR10 test images that were classified incorrectly when running AutoAttack (Croce & Hein, 2020) on our SOTA model. We performed classification of the adversarial images using (1) three human users and (2) GPT-4 (zero-shot classification). Each user was instructed to classify each image and provide a confidence value of either low or high. Following classification of the adversarial images, the human users also classified and provided confidence ratings for the clean images. The three human users are all members of our research team. We chose GPT-4 as a fourth user to supplement our small-scale human study, as GPT-4 has strong object recognition abilities and can understand the task via instructions similar to those that we give humans (Yang et al., 2023). Additional experimental details including the GPT-4 prompt are available in Appendix H.

The human user performance on the clean and adversarial versions of the 2629 images misclassified by our SOTA NN is shown in Figure 14, showing that average attacked-data classification performance is 25% lower than average performance on the corresponding clean images. Indeed, every user's performance drops by at least 21% when they move from clean to attacked data, supporting the existence of a limit on robustness below 100% and thus also supporting our scaling laws' modeling of irreducible error on this problem.

Moreover, assuming the human users and GPT-4 were able to correctly classify the 7371 adversarial images that our SOTA model classified correctly, then the average human accuracy would be 90.46% on all of the adversarial images – for context, GPT-4 would have an accuracy of $85.31\%$ as it obtained 44.12% accuracy on the 2629 adversarial images.[4] Interestingly, the limits indicated by our scaling laws and the limits indicated by our human study are consistent, as the average human user accuracy of 90.46% closely aligns with the asymptotic adversarial CIFAR10 test accuracy for FID=0 predicted by our scaling law (we illustrate the closeness of these estimated performances in Figure 1).

### G.2. Addressing Invalid Data

The inability of human users to correctly classify $\sim 10\%$ of adversarial images suggests that the $l_\infty$ attacks performed on our SOTA model produce invalid images, lacking a perceptible depiction of the labeled class. Naively, we could define invalid images as the set of images that are never correctly guessed, but this would not reflect that some guesses are correct by chance. Indeed, we find that humans who lack confidence in recognizing an instance of the labeled class in the image are often correct nevertheless (see Figure 14). Thus, to reduce the extent to which we label as "valid" the images that are

---

[4]These numbers do not represent how users would perform if they were adversarially attacked, but rather indicate the extent to which the correct classes are visible in the images that were designed to (and successfully did) attack our SOTA model.

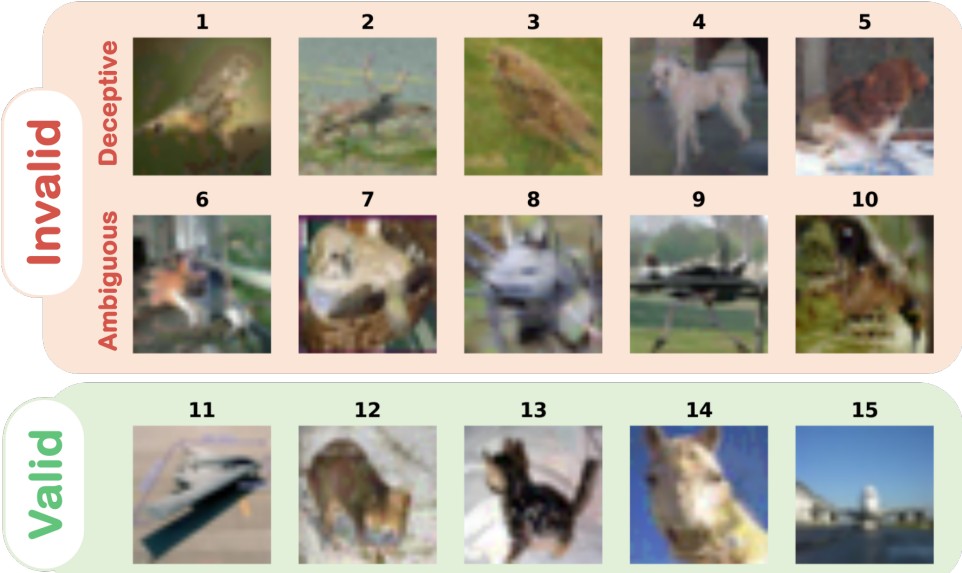

*Figure 15.* **A sampling of valid and invalid adversarial images.** We categorized 727 out of 2629 adversarial images as invalid based on our human and GPT-4 study. Invalid images revealed unanticipated behaviour of adversarial attacks, adding to images instances of objects with incorrect labels ("Deceptive") and making the labeled instance imperceptible ("Ambiguous"). The ground truth classes are below[6]. Clean images are provided in Appendix H.

*Table 4.* Benchmarking has a new performance limit when it excludes invalid data. Our SOTA NN's performance with and without invalid data is shown in the first row. We also estimate upper bounds on average human performance: the first column of the second row adds the 7371 images correctly classified by the SOTA NN to the average number of images correctly classified by humans when they are given the 2629 images misclassified by our SOTA NN, then divides by 10000. Removing invalid data moves the estimated average human performance closer to 100%.

| Configuration | All Data | Valid Data |
|---|---|---|
| SOTA NN | 73.71% | 79.49% |
| SOTA NN + Avg Human | 90.46% | 96.46% |

correctly guessed by chance, we define valid images as the $1902/2629$ images correctly labeled by $2/4$ users or 1 *human* user with high confidence—users with high confidence are typically correct and likely see an instance of the label class in the image (Figure 14). The other $727/2629$ images are categorized as invalid.

In Figure 15, we show examples of **valid** and **invalid** images. We also illustrate two flavors of invalid images: (i) *deceptive* images that depict an object with an incorrect class, deceiving users; and (ii) *ambiguous* images that depict no object clearly, confusing users. *Deceptive* images, representing 39% of invalid images, are those causing our SOTA NN and a majority of humans to make the same incorrect prediction.

Interestingly, prior work shows that visually ambiguous or illusory images can be interpreted in multiple ways by both humans and machines, e.g. depending on the prompt given to a generative classifier (Jaini et al., 2023). Uniquely, we show that data generated by adversarial attacks harms performance not just by creating ambiguity that classifiers fail to deal with, but rather it can also create a clear depiction of an incorrect class that deceives both human and machine users into confidently making the same incorrect prediction. Indeed, visual inspection of the *deceptive* images, which make up 39% of all invalid images, illustrates clear depictions of incorrect classes (e.g., see the images in the top row of Figure 15 and the right side of Figure 1).

To better track robustness as a fraction of human performance, we remove from the CIFAR10 $\ell_\infty$ adversarial benchmark test

---

[6]1:cat, 2:bird, 3:deer, 4:horse, 5:deer, 6:cat, 7:dog, 8:bird, 9:bird, 10:cat, 11:airplane, 12:deer, 13:cat, 14:horse, 15:airplane

set (Croce et al., 2020) the images that AutoAttack can make invalid. Table 4 shows our SOTA NN's robustness on the new and original benchmarks. Clarifying the limitations imposed by invalid images, Table 4 also shows the SOTA NN's 7371 correct classifications plus the average user's correct classifications on the 1902 (2629) valid (original) images remaining, divided by 9273 ($10^4$).

### G.3. Limitations

Note that our human study was limited to three users. To overcome this limitation, we plan to make our adversarial data publicly available in a quiz format, allowing the community to contribute to the estimates we made in Section G. Finally, while we provide an updated, more optimistic view of progress towards human-level robustness on CIFAR10, our scaling laws also suggest that the costs of reaching human performance are impractically high. More efficient algorithms (Bartoldson et al., 2023) for adversarial training can address these costs and open up paths to robust models.

## H. Invalid Adversarial Data Details

### H.1. Human Evaluation

To perform the human evaluation, we created a web application (see Figure 16). Users classified images in batches of 100, at their convenience in their own homes/offices over several days, and a unique web page was used for each batch of 100 images. Users classified the 2629 images that AutoAttack created which successfully tricked our SOTA NN, and they classified the clean versions of these images (thus, a total of 5258 images were classified by each of the 3 users).

The participants in our study were members of our research group. Unlike Shankar et al. (2020), we did not give users extensive training before they participated in this study. Being members of our research group, the users were familiar with CIFAR10. Further, CIFAR10 only has 10 classes, all of which are relatively distinct, reducing the importance of providing training/instruction beyond that which appears in the web application (shown in Figure 16).

### H.2. GPT-4 Evaluation

For completeness, we provide the prompt used for GPT-4 evaluation of adversarial images in Figure 17. Additionally, we provide the payload used for GPT-4 in Figure 18. It is important to note that while GPT-4 is able to accurately correctly classify 1160 out of the 2629 adversarial images, this does not imply that GPT-4 is more adversarially robust than our SOTA model on the CIFAR10 $\ell_\infty$ benchmark, as the adversarial images evaluated by GPT-4 we generated via attacks on our SOTA model. GPT-4 was selected as a fourth used for evaluation of the adversarial images due to its ability to understand subtle image details and its ability to understand the task via prompting Yang et al. (2023). However, this evaluation of GPT-4 on these adversarial images does suggest that adversarial attacks on the WideResNet-94-16 architecture, used for our SOTA model, are not fully transferable to GPT-4.

Trying numerous prompts would facilitate the optimization of GPT-4V performance and thus a potentially better understanding of which images truly have the information about the label destroyed by adversarial attacks. Prior to settling on the prompt we chose, we conducted a preliminary study that explored on a subset of images a prompt that expressed to GPT-4V that some images may have been attacked (and how that might affect object appearances), which led to a small improvement in performance (~5%). Given the marginal improvement in performance and the significantly enhanced complexity of the prompt (that provided details the human users were not explicitly given), we opted for the prompt shown in Figure 17.

### H.3. Clean versions of Figure 15

In Figure 19, we provide the clean (i.e., unattacked) versions of the images provided in Figure 15 with the same invalid and valid categorization. Of note, the classes of the images in the "Deceptive" subcategory are much clearer and very different than the perceived classes by humans and our SOTA NN. Specifically, perturbations in the "Deceptive" adversarial images in Figure 15 appear to produce attributes of birds, deer, or dogs that are clearly not present in the original images. Additionally, humans found the perturbations that were applied to the "Ambiguous" adversarial version of the images made them appear warped or imperceptible while the clean images were easier to interpret and classify.

**Guess the perturbed image!**
**Instructions:**
- For the first image, guess a label and choose a confidence in your guess: if it is very hard to deduce the label then choose 'Low', otherwise choose 'Higher'.
- Repeat this for the second image, it will appear after you finish the first.
- **IMPORTANT:** 'Automobile' includes sedans, SUVs, etc. 'Truck' includes only big trucks. Neither includes pickup trucks.
- Do not refresh your browser, you'll lose your progress!
- When you are done, click the 'Save to CSV' button and send us the generated file.
**TLDR: choose a confidence and label for the first image then the second image to finish the tab and see your performance.**

| Tab 0 | Tab 1 | Tab 2 | Tab 3 | Tab 4 | Tab 5 | Tab 6 | Tab 7 | Tab 8 | Tab 9 | Tab 10 | Tab 11 | Tab 12 | Tab 13 | Tab 14 | Tab 15 | Tab 16 | Tab 17 | Tab 18 | Tab 19 | Tab 20 | Tab 21 |
| Tab 22 | Tab 23 | Tab 24 | Tab 25 | Tab 26 | Tab 27 | Tab 28 | Tab 29 | Tab 30 | Tab 31 | Tab 32 | Tab 33 | Tab 34 | Tab 35 | Tab 36 | Tab 37 | Tab 38 | Tab 39 | Tab 40 | Tab 41 |
| Tab 42 | Tab 43 | Tab 44 | Tab 45 | Tab 46 | Tab 47 | Tab 48 | Tab 49 | Tab 50 | Tab 51 | Tab 52 | Tab 53 | Tab 54 | Tab 55 | Tab 56 | Tab 57 | Tab 58 | Tab 59 | Tab 60 | Tab 61 |
| Tab 62 | Tab 63 | Tab 64 | Tab 65 | Tab 66 | Tab 67 | Tab 68 | Tab 69 | Tab 70 | Tab 71 | Tab 72 | Tab 73 | Tab 74 | Tab 75 | Tab 76 | Tab 77 | Tab 78 | Tab 79 | Tab 80 | Tab 81 |
| Tab 82 | Tab 83 | Tab 84 | Tab 85 | Tab 86 | Tab 87 | Tab 88 | Tab 89 | Tab 90 | Tab 91 | Tab 92 | Tab 93 | Tab 94 | Tab 95 | Tab 96 | Tab 97 | Tab 98 | Tab 99 |

Your perturbed image robustness is 0% (0 out of 0) and alignment with NN is 0% (0 out of 0)

Your original image accuracy is 0% (0 out of 0) and alignment with NN is 0% (0 out of 0)

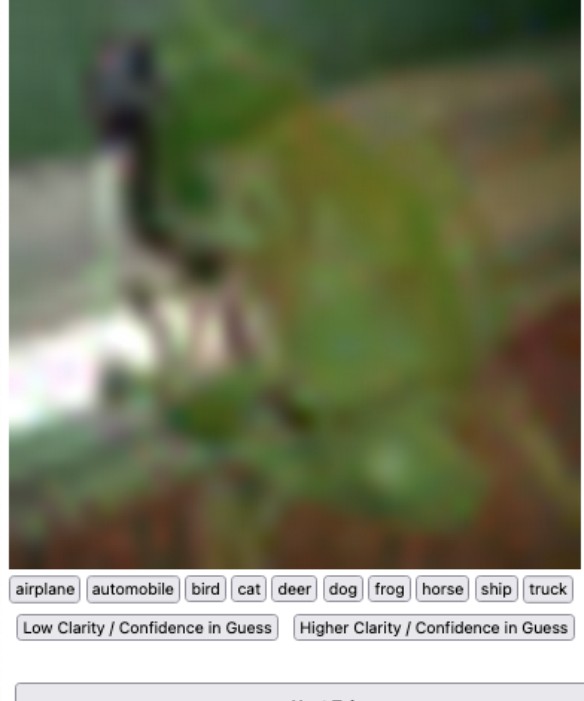

| airplane | automobile | bird | cat | deer | dog | frog | horse | ship | truck |

| Low Clarity / Confidence in Guess | Higher Clarity / Confidence in Guess |

| Next Tab | Save to CSV |

*Figure 16.* The web application used to collect data in our human evaluations. Users were given instructions verbally, which were repeated at the top of the web page.

```
prompt = ("I will give you an image that contains one of 10 classes. "
          "The 10 classes are [airplane, automobile (but not truck
           or pickup truck), bird, cat, deer, dog, frog, horse,
           ship, and truck (but not pickup truck)]. "
          "Your reply to the image will tell me which of the 10
           classes the image contains. "
          "Feel free to reason about the contents of the image, but the
           final two words of your response must be in the following
           format: '{class}, {confidence}'. "
          "'class' must be one of the 10 classes. 'confidence' must be
           'low' or 'higher' -- only say 'low' if you think it is
           unlikely that you have identified the correct class.")
```

*Figure 17.* GPT-4 Prompt for Evaluating Adversarial Images.

```
payload = {
    "model": "gpt-4-vision-preview",
    "messages": [
      {
        "role": "user",
        "content": [
          {
            "type": "text",
            "text": prompt
          },
          {
            "type": "image_url",
            "image_url": {
              "url": img_base64
            }
          }
        ]
      }
    ],
    "max_tokens": 3200,
    "temperature": temperature
}
```

*Figure 18.* GPT-4 Payload for Evaluating Adversarial Images. "temperature" is set to 0, and "img_base64" is the Base64 representation of the image.

### H.4. Adversarial Training Images Can Be Invalid

The ability of benchmark attacks to make 10% of CIFAR10 adversarial test images invalid suggests that weaker, training-time attacks may generate invalid data too. In Figure 20, we show the original and AutoAttack-perturbed images from Figure 1 alongside the original image after the 10-step PGD attack used for training. Interestingly, the training attack results in an image (Figure 20, middle image) that appears to humans to be either a dog or a horse, or an interpolation between the two. The existence of invalid training images suggests that invalid adversarial data could also be removed at training time, rather than just during benchmarking.

### H.5. Adversarial Images That Can Fool Humans

In Figure 21, we shown adversarial images that can potentially fool humans along with their clean counterparts.

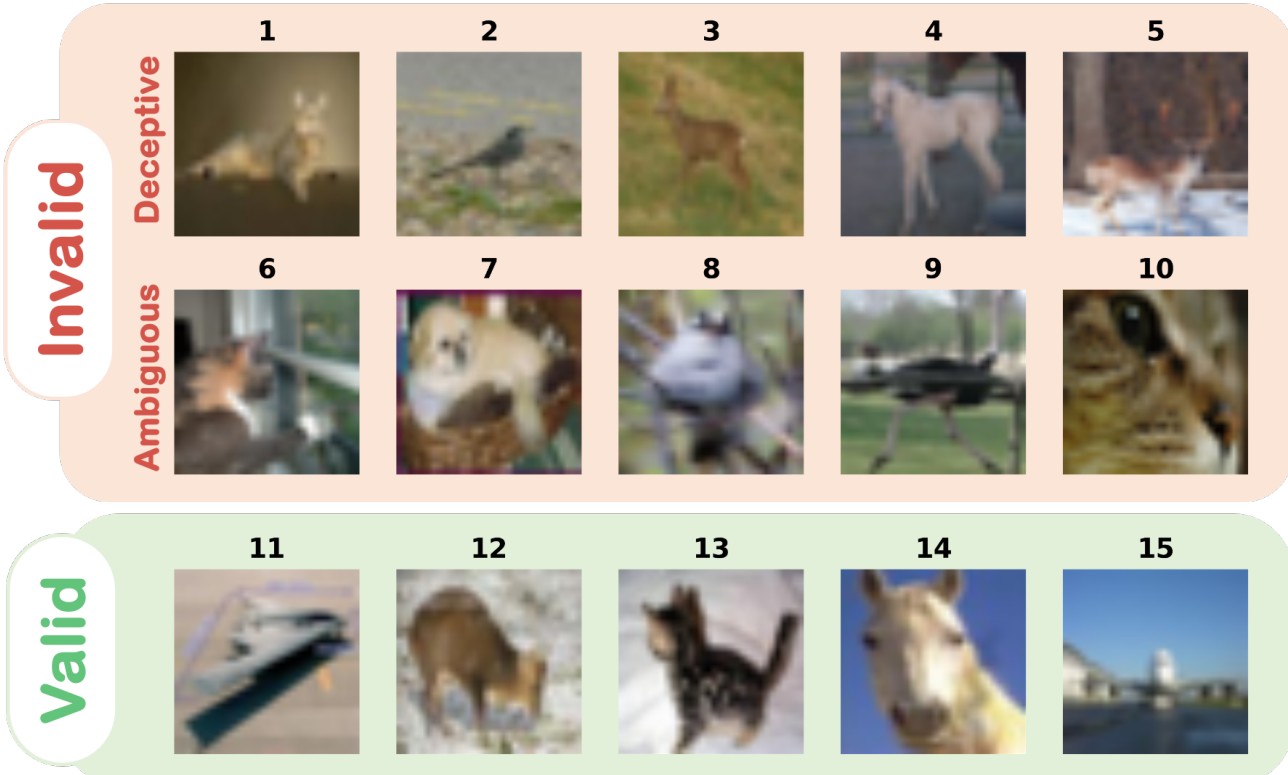

*Figure 19.* **Clean versions of adversarial images from Figure 15.** We categorized 727 of the 2629 adversarial images as invalid (27.87% of images) based on analysis of our human and GPT-4 evaluations. Here we provide the original clean versions of the adversarial images from Figure 15 with the same categorization and numbering. The ground truth classes are: 1:cat, 2:bird, 3:deer, 4:horse, 5:deer, 6:cat, 7:dog, 8:bird, 9:bird, 10:cat, 11:airplane, 12:deer, 13:cat, 14:horse, 15:airplane. The NN predicted the following classes on the adversarial versions of these images: 1:bird, 2:deer, 3:bird, 4:dog, 5:dog, 6:deer, 7:frog, 8:frog, 9:airplane, 10:deer, 11:ship, 12:cat, 13:dog, 14:cat, 15:ship.

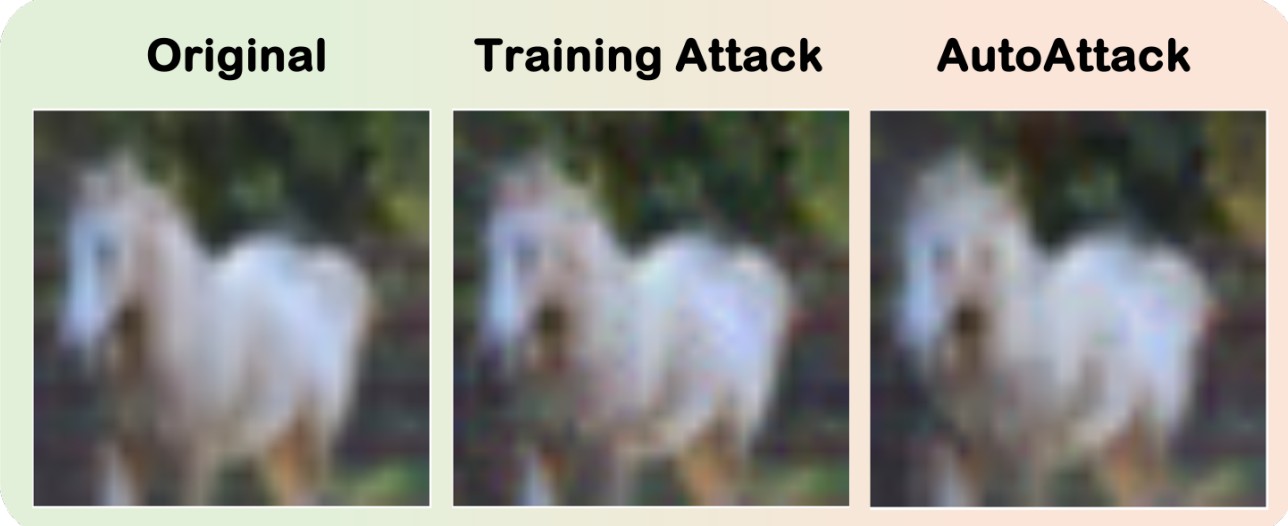

*Figure 20.* **Invalid training data.** While weaker than benchmark attacks, training attacks like 10-step PGD can produce adversarial data that fools humans. The middle image has the label "horse" but may appear as a "dog".

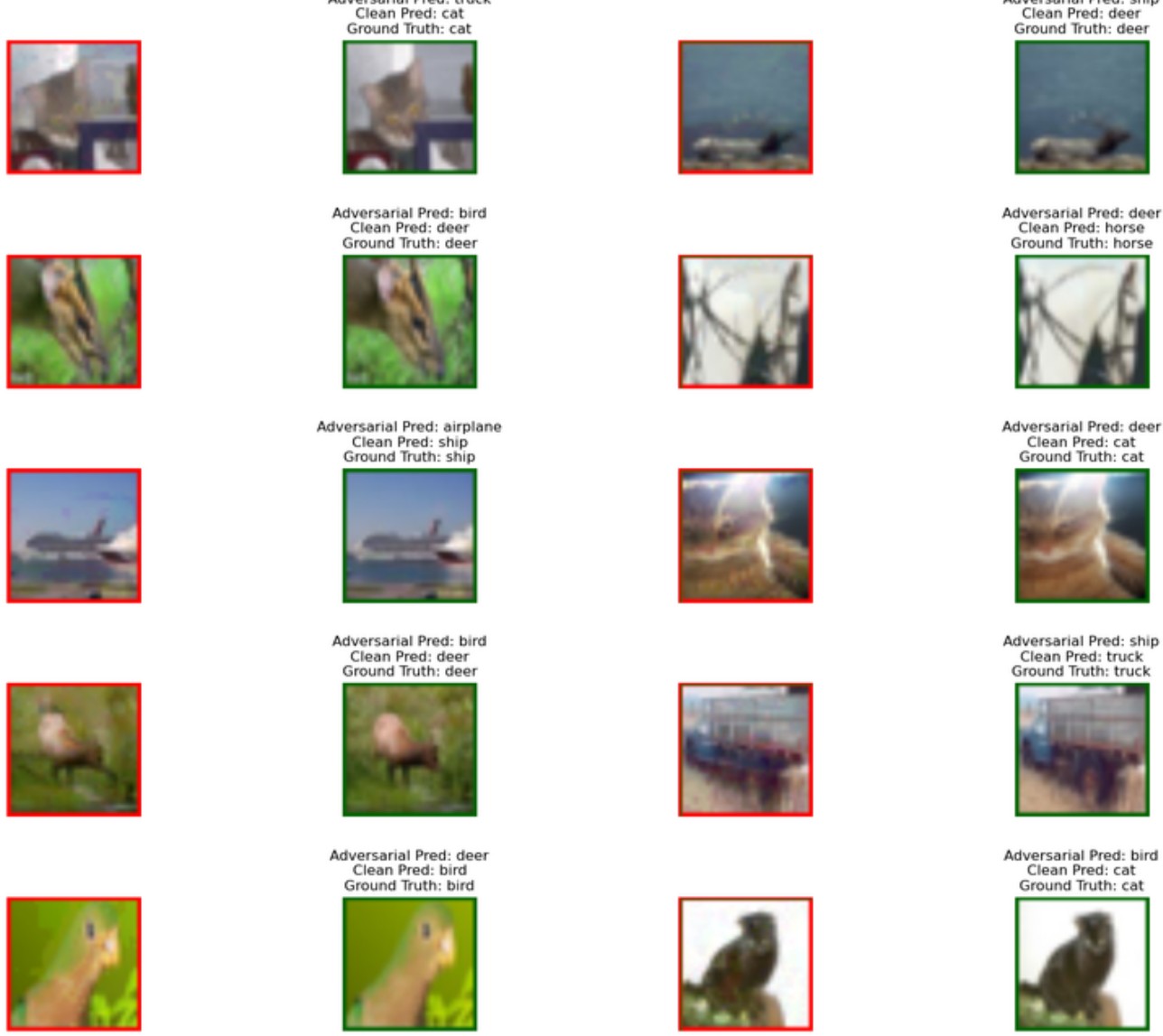

*Figure 21.* **Sophisticated manipulation strategies emerge with large-scale adversarial training, even when using naive attack algorithms.** Notice the subtle yet effective manipulations of clean images that can, in many cases, fool the human visual system. $\ell_\infty$ AutoAttack identifies the important parts of the image and then applies a combination of adversarial tactics, such as edge and texture tempering, discoloration, etc., to create effective adversarial examples. Adversarial examples from Figure 2 are bounded by red boxes and reproduced here for convenience, their clean/original versions are bounded by green boxes.

