# OpenReview forum: "Adversarial Robustness Limits via Scaling-Law and Human-Alignment Studies"
_ICML.cc/2024/Workshop/WANT — WANT@ICML 2024 Poster_

### Official Review · Reviewer_jbqs · 2024-06-16
**Useful scaling laws to evaluate compute efficiency and data quality's effect. The impact is however restricted to a specific problem on a single dataset.**

**Confidence:** 3

**Summary:**

This paper proposes scaling laws for the accuracy of image classifiers under adversarial attacks, depending on their number of parameters as well as the size and quality of the dataset used for training. It provides 3 approaches to find the optimal compute efficiency model for a task, which allows the authors to outperform the SOTA on AutoAttack accuracy while reducing the model's number of parameters.

They also show that classification robustness to adversarial attack scales logarithmically with FLOPs, restricting the achievable robustness to around 90% accuracy on this task. They argue that it is also the accuracy achieved by a human on the same problem.

**Strengths:**

- These novel scaling laws take into account data quality, which was not done before.
- The paper proposes a model that improves SOTA on AutoAttack classification task.
- Different experiments were conducted in various settings, showing consistent results.

**Weaknesses:**

- In the introduction & related work, the problem of *invalid data* and human robustness is presented as part of the work. However it is not explored in the main paper, only in the appendices.
- AutoAttack is the only task evaluated. Is there any other available evaluation for the robustness of your method ?
- L.320 : It is not clear to me where the $7822 N D$ FLOPs constraint comes from.
- No code provided for reproducibility.

**Limitations:**

The proposed scaling laws only apply to image classification under adversarial attacks.\
All experiments were done on CIFAR-10 with data augmentation. No other dataset was tested.

---

### Meta-Review · Area_Chair_NYbh · 2024-06-17

**Recommendation:** Accept (Poster)
**Confidence:** 4

**Metareview:**

The paper received a single review. Upon checking the paper, the AC agrees with the assessment, with the work containing novel components and strong results. Hence, the AC recommends for acceptance.

---

### Decision · Program_Chairs · 2024-06-17

**Decision:**

Accept (Poster)

**Comment:**

We thank the authors for their time and contribution to WANT and we are pleased to share that after the reviewing process the paper has been accepted. Congratulations! We encourage the authors to consider reviewers' feedback for the improvement of the camera-ready version. We hope to see you in person at the workshop and brainstorm on efficient training research together!